# Assisted Suicide in Austria: Nurses’ Understanding of Patients’ Requests and the Role of Patient Symptoms

**DOI:** 10.3390/ijerph22020218

**Published:** 2025-02-04

**Authors:** Matthias Unseld, Alexa L. Meyer, Tamina-Laetitia Vielgrader, Theresa Wagner, Dorothea König, Chiara Popinger, Bärbel Sturtzel, Gudrun Kreye, Elisabeth L. Zeilinger

**Affiliations:** 1Department of Clinical Research SBG, Academy for Ageing Research, Haus der Barmherzigkeit, A-1160 Vienna, Austria; matthias.unseld@hb.at (M.U.); alexaleonie.meyer@hb.at (A.L.M.); baerbel.sturtzel@hb.at (B.S.); 2Division of Palliative Medicine, Department of Medicine I, Medical University of Vienna, A-1090 Vienna, Austria; 3Institute for Ethics and Law in Medicine, University of Vienna, A-1090 Vienna, Austria; tamina-laetitia.vielgrader@univie.ac.at; 4Department of Clinical and Health Psychology, Faculty of Psychology, University of Vienna, A-1010 Vienna, Austria; theresa.wagner@univie.ac.at (T.W.); dorothea.koenig@univie.ac.at (D.K.); 5Clinical Division of Palliative Medicine, Department of Internal Medicine II, University Hospital Krems, Mitterweg 10, A-3500 Krems, Austria; gudrun.kreye@krems.lknoe.at; 6Karl Landsteiner University of Health Sciences, A-3500 Krems, Austria

**Keywords:** medical assistance in dying, nurses, end-of-life care, palliative care, hospice care, psychosocial symptoms, ethics, healthcare legislation, decision making

## Abstract

This study explores Austrian palliative and hospice care nurses’ experiences regarding assisted suicide (AS). Following its legalization in 2022, occupational groups affected by the legislation, such as nurses, have been left without clear guidance or instructions on how to navigate this new landscape. This study aimed to explore how nurses perceive their patients’ desire to die and its connection to the symptoms experienced by the patients. A cross-sectional online questionnaire survey was disseminated to all palliative and hospice care facilities in Austria and was eventually completed by 145 nurses, focusing on their understanding of patients’ requests for AS and the severity of patients’ symptoms. Factor analysis was used to identify symptom clusters, and Spearman rank correlations were employed to explore associations between nurses’ understanding of AS and factors such as patient symptoms, nurse demographics, and attitudes toward AS. The results indicate that psychosocial factors, particularly loss of dignity (63.6%) and autonomy (76.4%), were the most frequently reported severe symptoms. Understanding patients’ decisions was significantly associated with nurses’ general support for AS (*r* = 0.34, *p* < 0.001) but not with age, work experience, or religious beliefs. Factor analysis revealed four symptom clusters, with ’loss of dignity’ showing a small but significant correlation with nurses’ understanding of patients’ requests (*r* = 0.17, *p* = 0.044). The present findings highlight the importance of integrating psychosocial support into palliative care and emphasize the need for clear guidelines and training to better support nurses in managing AS-related challenges.

## 1. Introduction

The debate surrounding the right of terminally ill patients to voluntarily end their lives has gained significant attention in recent years. While dying with dignity and without suffering is widely regarded as a fundamental human right, voluntary assisted dying remains legalized in only a limited number of countries, despite considerable public support [1,2]. Currently, this includes eight European countries (the Netherlands, Belgium, Luxembourg, Switzerland, Germany, Austria, Spain, and Portugal), as well as Canada, Colombia, New Zealand, several states in Australia, and eleven U.S. states (California, Colorado, District of Columbia, Hawaii, Maine, Montana, New Mexico, New Jersey, Oregon, Vermont, and Washington) [3]. Other countries, such as France, Italy, and the UK, are also considering legalizing voluntary assisted dying [3,4]. In most cases, legalization applies to assisted suicide (AS), where medication is provided for self-administration, while euthanasia—where a healthcare professional administers the medication—is only legal in the Netherlands, Belgium, and Luxembourg. In all cases, voluntariness is, among other aspects, a key requirement, with general eligibility often being based on the presence of a serious and incurable illness.

In Austria, the ban on assisted suicide was lifted by the Constitutional Court in 2020, and the Assisted Dying Act (Sterbeverfügungsgesetz, StVfG) came into effect in January 2022 [5]. The law permits AS for individuals who are of legal age, capable of making their own decisions, and suffering from a permanent, incurable, or terminal illness that causes unbearable symptoms. Applicants must personally set up a legally binding “Sterbeverfügung” (will to die), declaring their wish to end their life. To ensure free and self-determined volition, applicants must be informed about the process by two physicians, one of whom must have qualifications in Palliative Care (PC). These physicians also have to confirm that the patient is suffering from a terminal illness or chronic disease and that he or she is capable of making their own decisions. A mandatory 12-week waiting period is enforced to ensure the decision is well-considered, though this can be shortened to two weeks for those in the final stages of illness. Patients must administer the lethal drug themselves. Caregivers’ participation in the entire process of AS is strictly voluntary, enabling institutions to opt out of staff involvement in AS [5].

The introduction of AS presents healthcare workers with new responsibilities and ethical challenges. Experience in other countries where assisted suicide is legal reveals that nurses often feel left alone in the context of assisted suicide. On the other hand, they tend to be the first among healthcare personnel to be addressed by patients with a wish to end their life [6,7,8,9]. However, the role of nurses and their involvement in the assisted dying process is not clearly defined in Austrian law. Although physicians and pharmacists are referenced in the law, the specific designation of nurses is not specified. Consequently, nurses fall under the term “assisting person”, if applicable. This implies that, if care personnel are involved, they must be identified in the dying decree as “assisting persons” [10].

Attitudes toward AS among healthcare professionals vary, with studies suggesting that support is generally lower among healthcare workers than the general public [2,11], and especially low among healthcare professionals qualified in palliative care [8,12]. In a survey from Austria prior to AS legalization, 38.5% of the participating physicians were in favor of maintaining the prohibition of AS, and 51.3% stated that they would not perform it under any circumstances. In addition, more than half of respondents would feel legally insecure about providing AS, even if they were doing so within the legal framework [13].

In a Swedish study, while 41% of the interviewed physicians supported the legalization of AS, only 29% expressed their willingness to perform the measure themselves [14]. Similar trends have been observed in other countries [12,15]. Nurses tend to be less supportive of assisted suicide than physicians although attitudes vary widely and tend to be more positive in more recent studies [8,12,16,17].

Nurses’ ability to understand patients’ wishes for assisted suicide is crucial since it not only enhances compassionate care but also helps nurses manage their own emotional responses and ethical concerns surrounding assisted suicide [18]. Understanding the wish for assisted suicide can be shaped by several factors, including professional experience, personal values, religious beliefs, and level of exposure to end-of-life care. However, this understanding may also be influenced by the presence of symptoms in the patient, such as severe physical or psychosocial problems [19,20]. Common physical symptoms of patients who express a desire for assisted suicide include severe pain, fatigue, and weakness, particularly in patients with terminal illnesses such as cancer or neurodegenerative diseases. However, research shows that psychosocial factors can also play a prominent role in motivating these requests [21,22].

Given the evolving legal landscape and the complex role of nurses in assisted dying, there is a need for further research on how nurses experience their involvement in AS. Therefore, the aims of this study are to explore nurses’ understanding of patients’ wishes to hasten death, including their involvement in the decision-making process. It additionally focuses on identifying the symptoms most commonly experienced by patients seeking assisted suicide, shedding light on the physical and psychological factors that contribute to such requests. In addition, this study will investigate the relationship between nurses’ ability to understand patients’ wishes to hasten death and patients’ symptoms, as well as nurses’ characteristics. This multi-faceted approach aims to provide a deeper understanding of the dynamics between patients’ end-of-life wishes and nurses’ roles.

## 2. Materials and Methods

### 2.1. Study Design and Questionnaire

This study used a cross-sectional design and was based on an online questionnaire with the following content: The first section included questions on socio-demographic data and a question on the nurses’ general attitude towards AS and the stance of the employer. The second section was only administered to participants who indicated that they had previously cared for a patient who had requested AS. In relation to this patient, nurses were asked about their sense of involvement in the patient’s decision-making process and the comprehensibility of the patient’s decision. Additionally, participating nurses were asked to evaluate the severity of symptoms experienced by this patient who had requested AS using the Minimal Documentation System (MIDOS) [23], a German translation of the Edmonton Symptom Assessment Scale (ESAS) [24]. The ESAS was administered as part of the online questionnaire. No official patient records were used. The ESAS comprises ten common symptoms observed in patients receiving palliative care, namely, anorexia, depression, nausea, emesis, dyspnea, tiredness, pain, weakness, constipation, and anxiety/uncertainty. The severity of these symptoms is rated on a five-point scale. The list was complemented by six additional items: bedriddenness, insomnia, loss of dignity, loss of meaning in life, physical symptom burden/body image issues, and loss of autonomy. Further sections of the questionnaire investigated nurses’ attitudes towards AS more deeply and identified their support needs. These data are reported elsewhere [25,26].

Prior to data collection, the questionnaire was pre-tested on a sample of medical and nursing staff from a palliative setting, as well as psychologists and psychometricians. Following this, the questionnaire was adapted and refined.

### 2.2. Study Procedure

The survey was conducted online via the SoSci Survey platform from the end of September to the end of December 2022, with the intention of capturing nurses’ views after approximately one year of implementation of the new law. All relevant aspects of data protection and data security were observed throughout the survey. A link to the survey website was e-mailed to all palliative and hospice care facilities in Austria. A total of 225 facilities were contacted, including both inpatient and outpatient settings. The contact details of these institutions were obtained from the Austrian palliative and hospice parent organization “Hospiz Austria”.

Inclusion criteria for the present study were as follows: (1) being a caregiver in a palliative care or hospice setting in Austria, (2) having cared for at least one patient with a wish for assisted suicide, (3) having a minimum age of 18 years, (4) being able to give informed consent, and (5) sufficient knowledge of the German language to complete the questionnaire. Following the provision of study information, participants gave their informed consent on the initial page of the online questionnaire. This was achieved by clicking on a button to indicate agreement with the declaration of consent. The entire survey was conducted anonymously, with no identifying information collected. This ensured that the data could not be used to directly or indirectly identify a participant or their employer. This study was conducted in accordance with the Declaration of Helsinki and was approved by the Ethical Committee of the Medical University of Vienna (Nr. 1373/2022).

### 2.3. Data Analysis

The data were exported from the online platform, checked for plausibility, and cleaned as appropriate. As patient symptoms were the primary outcome measure, only data from participants who provided information on these symptoms were included in the data analysis, with a maximum of one missing item allowed. For descriptive statistics frequencies and percentages were employed. Due to the ordinal level of the data, Spearman rank correlation was used to test for associations between nurses’ understanding of patients’ wishes for AS and the following aspects: nurses’ age, length of time working in palliative care, importance of religion/spirituality, frequency of discussion with patients about their decision, nurses’ general attitudes towards AS, and patients’ symptoms and symptom clusters according to factor analysis. The significance level was set to 5%. As this was an exploratory study, no correction of the alpha level for multiple testing was applied. We used Cohen’s guidelines to interpret effect sizes [27].

Exploratory factor analysis was used to group the symptoms into clusters and, consequently, examine the associations between these clusters of symptoms and the comprehensibility of the patients’ wish to die. Factors were included if their eigenvalue was >1 and then subjected to a varimax rotation. Data were analyzed with SPSS Statistics for Windows, Version 29.

## 3. Results

### 3.1. Description of the Sample

A total of 170 respondents met the inclusion criteria. Of these, 145 had no or one missing item on patient symptoms and therefore represented the final sample for data analysis. This included 136 nurses who completed all symptoms for one of their patients and nine nurses who had one missing item. The mean age of the participants was 45.4 years (SD = 9.36) with *n* = 120 (82.7%) identifying as female. Most of the respondents had attained at least an upper-secondary education, and the majority were graduate nurses.

The median length of experience in palliative care among respondents was nine years. Almost two-thirds (64.6%) of the caregivers considered religion and spiritual aspects as rather important or very important in their lives, whereas only 4 (2.8%) stated that they were not important at all. While half of the respondents (*n* = 73; 50.3%) expressed at least some degree of support for AS, one in three did not approve of the new legislation (see Figure 1). More than half of the respondents (85; 58.6%) stated that their employer did not allow AS in their institutions, while 22 (15.2 %) said that it was permitted and 31 (21.4%) that it was either not clearly regulated, had not been communicated, or that they did not know. Sample characteristics are depicted in Table 1.

### 3.2. Understanding of the Patient’s Decision and Involvement of Nurses

Most of the respondents (*n* = 122; 84.7%) could at least somewhat understand the patient’s decision for wanting to hasten death. Only two participants (1.4%) could not understand this decision at all (see Figure 2). After the patient had expressed the wish for AS, about a quarter of the participants (25.5%) had frequently or very frequently discussed with the patient about their wish while another quarter (26.2%) had never or rarely done so, and 37.2% had done so occasionally. Most of the participants (61.4%) expressed no doubts or regrets about the way they discussed the matter with the patient. Only 15 (10.3%) of the nurses believed that their personal attitude about AS had influenced the patient and the majority (62%) did not feel involved in the decision of the patient to make use of AS. Only 6.2% felt rather or strongly involved in this decision (see Figure 2).

Understanding the patient’s reasons for wanting to die was not significantly associated with age, length of time working in the palliative care environment, the importance of religion and spirituality, or frequency of discussion with patients about their decision. On the other hand, understanding the patient’s reasons for wanting to die was positively correlated with the overall attitude towards AS (r = 0.344, *p* < 0.001, medium effect). The more the participants generally supported AS, the more they were able to understand the patient’s decision for wanting to die.

### 3.3. Symptoms of Patients Wishing for AS

Among patients’ symptoms, loss of autonomy and anxiety/uncertainty were most frequently reported as severe or very severe (76.4% and 66.9%, respectively). Other symptoms frequently reported as severe or very severe were loss of dignity (63.6%), weakness (63.4%), and loss of meaning (61.3%). About half of the patients experienced severe or very severe pain (55.2%), body image issues (50.3%), depression (45.5%), and tiredness (44.1%). Gastrointestinal symptoms, nausea, and insomnia were the least frequently reported symptoms (see Figure 3).

A significant correlation between individual symptoms and the nurses’ ability to understand the patient’s decision for AS was observed only for bedriddenness (*r* = 0.164, *p* = 0.05) and loss of autonomy (*r* = 0.176, *p* = 0.035), both with small effect sizes.

Exploratory factor analysis involving all patient’s symptoms revealed four factors with eigenvalues > 1. After Varimax rotation, these factors were named tiredness (1), emesis (2), anxiety (3), and loss of dignity (4) according to their main components (see Table 2). A mean was calculated for each factor, resulting in four means for each respondent. Correlation analysis showed that only one of these factor scores was significantly correlated with nurses’ understanding of the patient’s decision for AS, namely ‘loss of dignity’ with a small effect (*r* = 0.17, *p* = 0.044).

## 4. Discussion

The present study is among the first in Austria to explore the experiences of palliative and hospice care nurses toward AS following its legalization in 2022. A notable proportion of nurses expressed support for AS. Nearly 50% of respondents supported AS to varying degrees, with 19% being strongly supportive. In contrast, only 9% were entirely opposed. This aligns with recent studies showing a growing acceptance of AS among healthcare professionals, particularly nurses [17].

A high proportion of respondents (84.7%) could understand the patients’ reasons for wanting to hasten death, and this was particularly true of nurses who expressed a positive attitude towards AS but was unrelated to nurses’ age, work experience, or religious and spiritual beliefs. However, religion is often cited as a significant factor in opposition to AS [8,10]. This suggests that, while personal beliefs may shape nurses’ ethical perspectives, they may not necessarily impede their ability to empathize with patients’ decisions. This is supported by a Californian study on the impact of hospice social workers’ beliefs, ethics, and values on their work with patients requesting AS, in which most participants saw no influence or interference of their religious beliefs in the provision of services to the patients [28]. The ability to understand the patient’s reasons for wanting to hasten death was found to be crucial for healthcare workers in maintaining a professional approach, as it enables them to accept and respect the patient’s decision, regardless of their personal attitudes towards AS [18]. The ability to understand is also an important prerequisite and facilitator of interaction with the patient [18,29], although it does not necessarily promote the will to actively perform AS [30].

Attitudes towards AS may also shift depending on the relationship with the patient. For example, in a Polish study, nurses were more opposed to euthanasia for close relatives than for themselves [31]. In turn, a systematic review found that relatives of terminally ill patients were often more supportive of AS than the patients themselves [32].

Although many nurses in this study had discussed AS with their patients, most felt detached from the decision-making process and did not believe their own views influenced patient choices. This detachment is also reflected in a study of North Carolina clinicians, where the majority reported feeling capable of remaining objective when discussing AS with patients [33]. A professional and detached approach, regardless of personal beliefs, is considered essential for caring for patients in a professional and responsible manner [34,35]. However, engaging in conversations about AS is also seen as a key opportunity to support patients dealing with suffering, uncertainty, and distress. Studies suggest that addressing unmet concerns may even dissuade some patients from pursuing AS. Despite this, nurses often hesitate to engage in such discussions due to emotional, moral, or professional reasons [30,34,36,37]. The variation in how often respondents in this study discussed AS with patients—coupled with the fact that 11% did not specify their level of involvement—may indicate some discomfort with the topic. It is noteworthy that, despite the legal status of AS, more than half of the respondents worked in institutions that did not allow its practice. This institutional attitude may contribute to increased uncertainty and ethical dilemmas among nurses [38,39]. Patients, in turn, often want better communication and discussion with their caregivers, and some studies have expressed frustration at the lack of this [37].

The motivation behind advocating for AS is typically rooted in the desire to alleviate suffering and respect patients’ autonomy and dignity, consistent with this study’s finding that symptoms such as bedriddenness, loss of autonomy, and loss of dignity were strongly correlated with nurses’ understanding of patients’ requests for AS. Along with pain and weakness, psychosocial symptoms such as anxiety, loss of dignity, loss of autonomy, loss of meaning in life, and depression were also the symptoms most commonly reported by nurses in the present study. In turn, physical symptoms other than pain and weakness were much less observed. This finding is consistent with other research on the primary motivations for requesting AS. Loss of dignity and autonomy, rather than physical pain, are cited as the predominant reasons for choosing AS by patients and healthcare providers [32,37,40,41,42,43,44,45,46,47]. For example, in Oregon, between 1998 and 2022, 90.3% of AS cases were motivated by loss of autonomy, and 71.7% by loss of dignity [48]. In another survey in Oregon, physicians described their patients requesting AS as independent, self-determined, and self-sufficient who did not want to be dependent on others and become a burden. Accordingly, fear of losing control of their lives, becoming dependent on others, and losing their dignity was a major motive for requesting AS [41]. Physical symptoms were reported less frequently, and even pain is generally not perceived to be more severe in patients wishing to die than in other terminally ill hospice patients [40]. Fear of future pain was often more important than actual pain, and effective pain control could lead some patients to reconsider their wish for AS [32,37,42]. Interestingly, in a Swiss study comparing the reasons given by patients for requesting assistance in dying with the reasons given by physicians for prescribing the lethal drug, while pain was the most common reason given by both groups, loss of dignity was cited by 38% of the deceased, but by physicians in only 6% of cases [44].

AS is not always sought with the intent to act on it. Rather, it may provide patients with a sense of control, serving as a psychological safeguard against anticipated suffering [37,48]. In this way, AS may also function as a means for patients to communicate their unmet emotional and existential needs to their caregivers and relatives [37,49,50]. Loss of dignity and autonomy is accompanied by a sense of being a burden, both to oneself and to others [51,52], and may create feelings of guilt and dependency that can become unbearable for the patient [37,53]. These feelings can lead to depression and loss of meaning in their life, which many of the nurses who participated in this survey observed in their patients requesting AS.

Addressing psychosocial symptoms is crucial in delivering comprehensive end-of-life care, as it acknowledges the significance of psychological, social, and spiritual well-being in reducing feelings of despair and hopelessness [37,47,54]. Psychological and social support can help patients cope with emotional and existential distress, thereby improving their overall quality of life. However, recognizing and managing these aspects can be difficult for nurses, as psychosocial needs are complex and highly individualized. Yet, integrating psychosocial support into palliative care can promote a greater sense of hope and dignity in patients as they approach the end of life, thus preventing the risk of despair leading to requests for assisted suicide.

The nurses who participated in our survey not only identified psychosocial symptoms in their patients requesting AS, but these symptoms were also positively associated with their understanding of this request. A number of surveys have reported that nurses can understand the motives of patients who wish to hasten death, namely, intense mental and physical suffering and that a high proportion would also be willing to participate in the process as long as it was legal and the process was clearly defined [55]. However, participation in AS, even when it was legal, was also perceived as emotionally demanding [56]. It was found that nurses generally felt that they should be involved in decisions about AS, but also that there should be more information, training, and support on the subject of medical assistance in dying and a clearer definition of their specific role in the process [17,25,57,58]. These findings, along with evidence from the present study, indicating that many nurses have discussed AS with their patients, underscore the importance of integrating AS into both education and ongoing professional practice. Given the current legal situation concerning AS in Austria and its operational and organizational challenges and predicaments, it is recommendable to incorporate AS training into nursing curricula and provide practical case-based training. Such initiatives could better prepare nurses to navigate the complex legal and ethical dilemmas associated with AS. Training sessions could include case studies, legal education, incorporation of PC and end-of-life ethics as well as workshops in interdisciplinary teams to develop guidelines and protocols. Encouraging (guided) self-reflection among nurses is also essential to help them manage potential stressors, such as responding to patient questions about their personal views on AS, and ensuring they feel confident, prepared, and supported in addressing this sensitive topic effectively.

The present study was a pilot study and the first to investigate the topic of assisted suicide in Austria after the implementation of the new law, and to explore potential issues in this context relevant to nurses. Future research could focus on longitudinal studies to examine how nurses’ attitudes and experiences of AS evolve as the practice becomes more established in Austria. In addition, it would be valuable to investigate the effectiveness of training programs in preparing nurses for the ethical and emotional challenges associated with AS. Further investigation of the role of psychosocial support in mitigating AS requests, as well as the use of interdisciplinary collaboration in managing these cases, could be of particular interest. Finally, research on institutional policies and their influence on nurses’ experiences and involvement with AS could shed light on potential areas for organizational improvement.

Palliative care and assisted dying are often seen as antagonistic approaches to end-of-life care, although health professionals involved in AS have argued against this view and opted for the integration of assisted dying into palliative care [59]. The general positive attitude of Austrian palliative care and hospice nurses towards AS and their understanding of patients’ motives found in the present study would support this view.

### Strengths and Limitations

The present study is one of the first studies in Austria following the new legislation on AS, and not only provides insight into Austrian nurses’ attitudes towards AS after its recent legalization but also relates them to the presence of symptoms in patients. This supports the identification of determinants of the ability to understand the patient’s decision to hasten death.

However, this study has a number of limitations. The sample size of 145 nurses represents only a small fraction of the total nursing population in Austria. In addition, some large hospital operators did not facilitate the distribution of the survey to their staff, even after repeated requests. This may have been due to uncertainty about this new issue on the part of the operators, who have not yet established clear guidelines for their staff. Collaboration with large healthcare organizations could have increased participation but was avoided to prevent significant delays in data collection.

Furthermore, it is important to note the limitations of this study due to its online survey design. Sampling bias is a possibility, as individuals without internet access or those less tech-savvy may have been excluded, which limits the generalizability of the results. In addition, online surveys are often subject to low response rates and self-selection bias, where only highly motivated participants respond, potentially skewing the data. In the present study, nurses with a greater interest in AS may have been more likely to complete our questionnaire. Nevertheless, our results were obtained from healthcare professionals from eight out of nine Austrian federal provinces and can, therefore, be considered sufficiently representative for a pilot study.

## 5. Conclusions

This study indicates that Austrian palliative care nurses generally hold a positive attitude toward AS and understand patients’ reasons for requesting it, particularly those related to psychosocial distress such as loss of dignity and autonomy. These symptoms were also the most frequently observed by nurses in patients asking for assisted death. These findings emphasize the need for holistic approaches that especially address the psychosocial well-being of patients, not just physical care. It also highlights the critical role of nurses in end-of-life care and underlines the importance of providing nurses with better support, training, and guidance in handling the challenges associated with AS.

## Figures and Tables

**Figure 1 ijerph-22-00218-f001:**
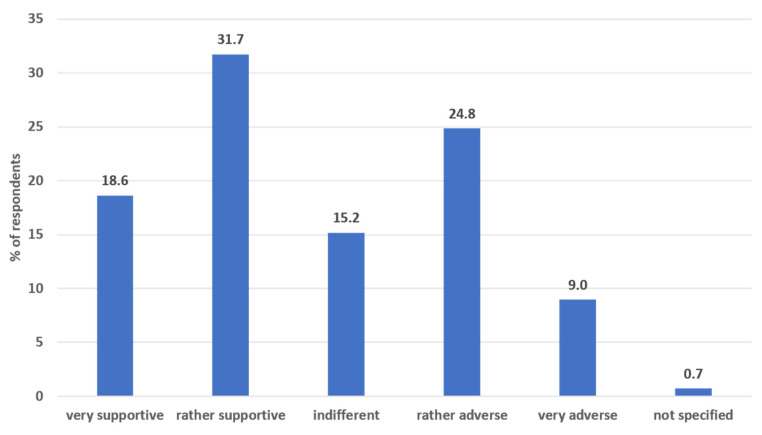
Nurses’ attitudes towards assisted suicide; figure caption: the figure shows the attitudes of nurses (*n* = 145) towards assisted suicide.

**Figure 2 ijerph-22-00218-f002:**
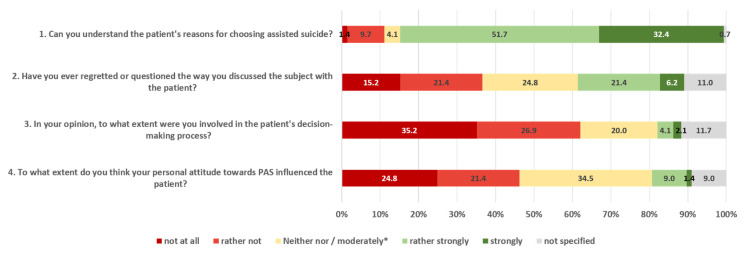
Understanding of the patient’s decision and experiences with AS; figure caption: the figure shows the distribution of responses to nurses’ experiences of a patient with a request for assisted suicide. * applies to question 4.

**Figure 3 ijerph-22-00218-f003:**
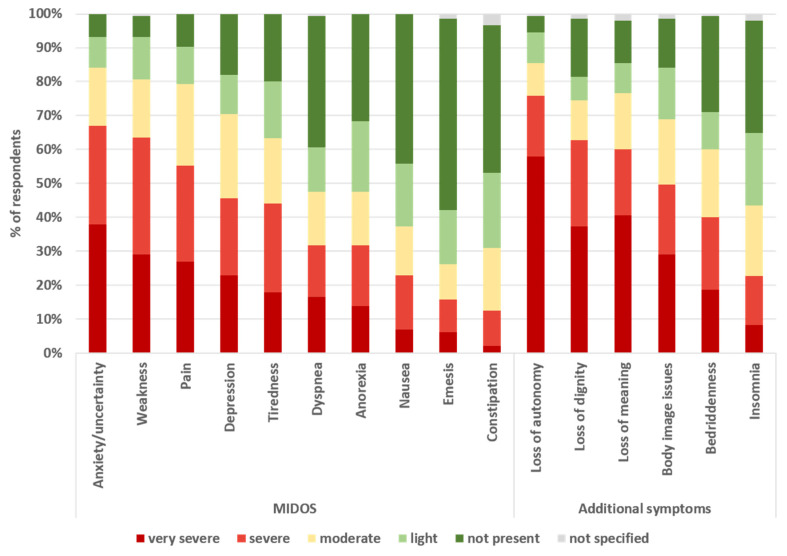
Occurrence of symptoms in patients requesting assisted suicide; figure caption: nurses (*n* = 145) were asked to rate the symptoms of the patient requesting assisted suicide on a 5-point scale. MIDOS: Minimal Documentation System, a German version of the Edmonton symptom assessment scale (ESAS).

**Table 1 ijerph-22-00218-t001:** Sociodemographic characteristics of the participants (*n* = 145).

	Frequency	Proportion, %
Sex (Female/Male/Non-Binary)	120/24/1	82.7/16.6/0.7
**Federal state**		
Lower Austria	44	30.3
Vienna	28	19.3
Tyrol	23	15.9
Upper Austria	17	11.7
Styria	17	11.7
Vorarlberg	9	6.2
Carinthia	5	3.4
Salzburg	1	0.7
Not specified	1	0.7
**Highest education level**		
Lower secondary (ISCED 1 + 2)	1	0.7
Vocational (ISCED 3)	10	6.9
Upper secondary (ISCED 3 + 4)	68	46.9
Bachelor (ISCED 6)	16	11.0
Master (ISCED 7)	35	24.1
Other	11	7.6
Not specified	4	2.8
**Occupational title**		
Graduated nurse (DGKP) ^a^	131	90.3
Specialized nursing assistant (level 2) ^b^	2	1.4
Nursing assistant (level 1) ^c^	6	4.1
Other	5	3.4
Not specified	1	0.7
**Duration of employment in palliative care in years** **(Median (IQR))**	9 (11.25)	

Note. IQR = Inter-quartile range; ISCED = International Standard Classification of Education ^a^ graduated from a school for nurses after 3 years; ^b^ graduated from a school for nurses after 2 years; ^c^ 1 year of education at a school for nurses or nursing assistance course.

**Table 2 ijerph-22-00218-t002:** Factor loadings of the symptoms.

Symptom/Component	Factor			
	1 Tiredness	2 Emesis	3 Anxiety	4 Loss of Dignity
MIDOS tiredness	0.792	0.062	0.249	0.044
MIDOS anorexia	0.678	0.262	0.079	0.074
MIDOS weakness	0.592	0.135	0.163	0.372
MIDOS constipation	0.414	0.188	0.291	0.095
Insomnia	0.398	0.229	0.355	0.087
MIDOS emesis	0.136	0.900	0.100	−0.026
MIDOS nausea	0.285	0.773	0.025	0.088
MIDOS dyspnea	0.123	0.446	0.117	0.297
MIDOS pain	0.168	0.347	0.366	0.280
MIDOS anxiety	0.115	0.083	0.679	0.141
Body image issues	0.114	0.092	0.607	0.244
MIDOS depression	0.240	0.108	0.496	0.088
Loss of meaning	0.166	−0.150	0.430	0.304
Loss of dignity	0.054	0.061	0.237	0.660
Loss of autonomy	0.051	0.044	0.209	0.642
Bedriddenness	0.281	0.255	0.075	0.487

Note. Extraction method: maximum likelihood; Varimax rotation. MIDOS = Minimal Documentation System. The background color indicates which symptom belongs to which cluster.

## Data Availability

The raw data supporting the conclusions of this article will be made available by the authors upon request.

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
