# Peer review of "Assisted Suicide in Austria: Nurses’ Understanding of Patients’ Requests and the Role of Patient Symptoms"

_ijerph, 2025, doi:10.3390/ijerph22020218_

Round 1
Reviewer 1 Report
Comments and Suggestions for Authors
Thank you for the opportunity to review this manuscript. The manuscript “Assisted suicide in Austria: Nurses’ understanding of patients’ requests and the role of patient symptoms” presents a well conducted cross -sectional study that investigates nurses understanding of patients' who wishes to hasten their death and also identifies the symptoms that are most commonly experienced by patients who seeking assisted suicide. Considering the broader context of presented studies and further impact of the published papers, I have couple of suggestions in the methodology and discussion section suggested to be addressed before next steps in the publication.
1. As mentioned in the methodology section, the survey link was sent via email to all palliative and hospice care facilities in Austria. However, no information is provided about the participation rate. Also, the sample size being low could be attributed to low participation? Being from hospital administration? Could be from the nurses? I would suggest including details about the number of contacted hospitals, number of hospitals agreed, number of nurses declined and so on, so that the readers understand the chance of participation bias if any. For this reason, a diagram could also be included.
Minor comments:
1. I would suggest replacing “Assisted suicide” in key words with a suitable similar word. As it has already been used in the title of the manuscript.
2. The authors have well presented strengths and limitations of the study. However, I would suggest including limitations associated with web-based questionnaire data also.
Author Response
Comments and Suggestions for Authors
Thank you for the opportunity to review this manuscript. The manuscript “Assisted suicide in Austria: Nurses’ understanding of patients’ requests and the role of patient symptoms” presents a well conducted cross -sectional study that investigates nurses understanding of patients' who wishes to hasten their death and also identifies the symptoms that are most commonly experienced by patients who seeking assisted suicide. Considering the broader context of presented studies and further impact of the published papers, I have couple of suggestions in the methodology and discussion section suggested to be addressed before next steps in the publication.
- As mentioned in the methodology section, the survey link was sent via email to all palliative and hospice care facilities in Austria. However, no information is provided about the participation rate. Also, the sample size being low could be attributed to low participation? Being from hospital administration? Could be from the nurses? I would suggest including details about the number of contacted hospitals, number of hospitals agreed, number of nurses declined and so on, so that the readers understand the chance of participation bias if any. For this reason, a diagram could also be included.
Thank you for your overall positive assessment of our manuscript. We have included more information about the number of organisations contacted under the headline “study procedure”, but as we also used a snowball technique, i.e. nurses were asked to share our survey link and invite other nurses, we cannot track how many nurses received the invitation to participate in the survey. As with all voluntary surveys, there is a selection bias. Nurses who are more interested in the topic of assisted suicide may have been more likely to participate in this study. In the revised manuscript, we have also made this point more explicit in the limitations section.
Minor comments:
- I would suggest replacing “Assisted suicide” in key words with a suitable similar word. As it has already been used in the title of the manuscript.
Thank you for this suggestion. We replaced “assisted suicide” with “medical assistance in dying”, a term that is particularly well known in Canada.
- The authors have well presented strengths and limitations of the study. However, I would suggest including limitations associated with web-based questionnaire data also.
We agree with the reviewer and have expanded our limitations section to include the shortcomings of online surveys.

Reviewer 2 Report
Comments and Suggestions for Authors
Dear Authors,
Thank you for your valuable contribution to the important and timely topic of assisted suicide.
My primary concern relates to the sample size. While this limitation has been acknowledged, I appreciate that your questionnaire includes participants from various provinces, which enhances generalizability despite the limited sample. However, you mention in the limitations that this is a pilot study. Could you please provide further clarification on this point? If so, could you please elaborate on the aims of the pilot study in relation to a potential larger study? Additionally, what are your plans for expanding this research in the future?
Also, you might want to take a step further and offer actionable recommendations for integrating AS into nursing education and practice, particularly in light of the legal and ethical challenges.
Author Response
Dear Authors,
Thank you for your valuable contribution to the important and timely topic of assisted suicide.
My primary concern relates to the sample size. While this limitation has been acknowledged, I appreciate that your questionnaire includes participants from various provinces, which enhances generalizability despite the limited sample. However, you mention in the limitations that this is a pilot study. Could you please provide further clarification on this point? If so, could you please elaborate on the aims of the pilot study in relation to a potential larger study? Additionally, what are your plans for expanding this research in the future?
Thank you very much for this valuable comment. This was the first study to look at the topic of assisted suicide in Austria after the implementation of the new law, so it was a pilot study to get an overview of potential issues in this context, relevant to nurses. In the revised paper we added the following plans for future research in the discussion section:
“The present study was a pilot study and the first to investigate the topic of assisted suicide in Austria after the implementation of the new law, and to explore potential issues in this context relevant to nurses. Future research could focus on longitudinal studies to examine how nurses' attitudes and experiences of AS evolve as the practice becomes more established in Austria. In addition, it would be valuable to investigate the effectiveness of training programs in preparing nurses for the ethical and emotional challenges associated with AS. Further investigation of the role of psychosocial support in mitigating AS requests, as well as the use of interdisciplinary collaboration in managing these cases, could be of particular interest. Finally, research on institutional policies and their influence on nurses' experiences and involvement with AS could shed light on potential areas for organizational improvement.”
Also, you might want to take a step further and offer actionable recommendations for integrating AS into nursing education and practice, particularly in light of the legal and ethical challenges.
We agree with the reviewer an added the following actionable recommendations in our discussion section: “Given the current legal situation concerning AS in Austria and its operational and organizational challenges and predicaments, it is recommendable to incorporate AS training into nursing curricula and provide practical case-based training. Such initiatives could better prepare nurses to navigate the complex legal and ethical dilemmas associated with AS. Training sessions could include case studies, legal education, incorporation of PC and end-of-life-ethics as well as workshops in interdisciplinary teams to develop guidelines and protocols.”
Round 2
Reviewer 1 Report
Comments and Suggestions for Authors
Thank you for the opportunity to review the revised version of the manuscript entitled “Assisted suicide in Austria: Nurses’ understanding of patients’ requests and the role of patient symptoms”. Based on the comments/suggestions and their implementations by the authors, the manuscript has improved manifolds. I have no further specific comments. I wish best of luck with the next steps in the publication. Have a successful 2025!